

# A database of water and heat observations over grassland in the north-east of Japan

Wenchao Ma[1], Jun Asanuma[1,2], Jianqing Xu[3], Yuichi Onda[1,2]

[1]Center for Research in Isotopes and Environmental Dynamics, University of Tsukuba, Tsukuba, 305-8577, Japan
[2]Faculty of Life and Environmental Sciences, University of Tsukuba, Tsukuba, 305-8577, Japan
[3]College of Business Administration, Kanto Gakuin University, Yokohama, 236-0037, Japan

*Correspondence to*: Wenchao Ma (wma@ied.tsukuba.ac.jp)

**Abstract.**  A highly valuable database of long-term hydro–meteorological measurements is presented, containing in
situ observations for a period of 37 years from well-maintained grassland in the north-east of Japan. The observations
include the shortwave radiation, net radiation, the air and dew point temperatures at three elevations, the soil temperature at
four depths, the sensible heat flux, soil heat flux, wind speed, relative humidity, air pressure and precipitation. The heights of
measurements are 1.6 m, 12.5 m and 29.5 m above ground, with the soil-layer observations at depths of −0.02 m, −0.1 m,
−0.5 m and −1 m. This high-quality database includes the four temporal resolutions of 10 s, 0.5 h, 1 h and 24 h, with the
hourly data presented here. The monthly and annual statistics are presented at the database web page of the Center for
Research in Isotopes and Environmental Dynamics and Prediction of the University of Tsukuba,
http://doi.org/10.24575/0001.198108. We have validated the data quality by comparing with published data from the local
meteorological agency in Tateno operated by the Japan Metrological Agency, including the average, maximum and
minimum values of air temperature, shortwave radiation, wind speed, relative humidity, and precipitation. We have
generated a daily downward longwave radiation time series with a method developed by Konto and Xu (1997) based on the
observations from the database. This constructed time series agrees well with observations collected between 2002 to 2006,
as evaluated based on the values of the Nash–Sutcliffe efficiency (=0.947) and the percent bias (=1.486). For the whole
database, annually averaged values give an obvious positive trend in the precipitation, air temperature, shortwave radiation,
net radiation and sensible heat flux over the previous 37 years, with a negative trend detected for the wind speed, soil heat
flux and soil temperature.

## 1 Introduction

In situ observational databases play a dominant role in all research disciplines. In hydro–meteorological studies, water
and temperature observations from field work are essential for validating differently sourced data (Jackson et al., 2009,
Liang, 2011, Guillevic et al., 2012), supporting theoretical modeling (Grayson and Bloschl, 2001), estimating the energy



budget (Hirschi and Seneviratne, 2017; Makarieva et al., 2017), and recording the historical climatic variation (Godsey et al., 2017, Kormos et al., 2017) for different subjects (Qu et al. 2016, Godsey et al., 2017). A full set of in situ hydro–meteorological observations is introduced here from well-maintained grassland located in the north-east of Japan, which has been continuously operated since 1981 by the Environmental Dynamics and Prediction (EDP) department of the Center for

Research in Isotopes and Environmental Dynamics of the University of Tsukuba, Japan. This observational site has a long history of providing important contributions to multidisciplinary investigations in many aspects of geological, hydrological, biological and meteorological studies, including micrometeorology both within and outside of the plant community, as well as investigations into the transportation of turbulence, evapotranspiration, the soil energy budget, and groundwater movement (Kawamura, 1991).

For decades, the EDP department has provided a high-quality database for many valuable studies assessing the energy balance and the degree of evapotranspiration. For example, Nakagawa (1983) tested the possibility for estimating the actual evapotranspiration using an equilibrium evaporation model. Sugita et al. (1985) developed an apparatus for measuring the heat pulse velocity for estimation of the transpiration flux by employing the energy-budget method, with the effect of the stemflow and vegetation storage on the evapotranspiration investigated. Sugita and Kotoda (1985) tested the effects of the

soil-water deficits on forest evapotranspiration based on the observations from the EDP database, from which the effective rate of soil-moisture consumption was estimated and shown to be more than half the amount controlling the suppression of the evapotranspiration. In that same year, another advanced technology was employed (Sugita and Kotoda, 1985), which combined using the remotely sensed, land-surface temperature, as well as Priestley–Taylor-type equations, for estimating the regional evapotranspiration. By assessing the observed evaporation in 2001, the results estimated from the Penman, energy-

budget eddy covariance, and energy-balance Bowen ratio methods were presented (Yubasaki et al., 2005), which improved understanding of the variation in evaporation from a conversion in the fraction of pasture at the site into turf. Saito and Yamanaka (2005) performed a quality control of the data, and analyzed the evapotranspiration data observed with a weighing Lysimeter between 1981 and 2002, with the results of the data quality summarized. The validation of the water budget was carried out by Tase and Majima (1985), where a model for the estimation of the precipitation on the grassland of

the EDP department was developed, showing a good adaptability with a model taking into account the canopy, stem and evapotranspiration components based on observations from the EDP database. The latent heat flux was also assessed by Hiyama et al. (1993), with the flux behavior compared before and after the precipitation event, as well as an investigation into the temporal variation, and the assessment of the measurement accuracy.

Investigations into the water and energy transfer, not only above the ground, but also within the soil layers, have also

been extensively carried out. For example, Sakura (1979) tested the infiltration effect for soil-water movement and soil temperature. Taniguchi (1990) specified the distribution of heat transport within soil layers by using observations of the soil temperature. Sakura and Taniguchi (1983) conducted experiments at the EDP site to test the characteristics of soil-water movement during infiltration. Measurement errors were investigated by Iwata and Sugita (2006), who demonstrated the reliability of the observed heat fluxes.



A variety of studies on the ecology and vegetation were broadly conducted because of the unique land surface covered by grassland. For example, Kotada and Hayashi (1980) have investigated the micrometeorology of the vegetation, with specifying empirical parameters describing the vegetation growth state based on physical modeling. While Nasuno et al. (1989) estimated the turbulent fluxes with the eddy-covariance method for assessing the effect of the forest and vegetation

on the exchange of energy and water vapor. Another study was carried out by Hayashi et al. (1989) to investigate the water vapor and temperature profile on this grassland. Being a well-maintained observation site, the grassland gives good conditions for fundamental ecological research. For example, Yasui and Oikawa (1993) investigated the $CO_2$ flux resulting from soil respiration, showing the high correlation between the soil temperature, moisture and respiration. Based on the unique land-surface characters, Yamanaka et al. (2005) conducted experiments for investigation of the biomass and root

characteristics at the site.

As a newly developed and reliable approach, the isotope method has also been applied extensively in the investigations of the water and temperature observations within the EDP database. Generally, branches of isotopic research in hydrological applications include the fields of hydro–meteorology, eco-hydrology, groundwater hydrology, watershed hydrology, and the use of isoscapes (Yamanaka, 2012), with isotopic techniques being well accepted as powerful and stable methods for

partitioning the transpiration from the evapotranspiration. In particular, Shimizu and Yamanaka (2005) tested the spatial structure of the isotopic composition of the atmospheric water vapor at the micrometeorological scale over this observational site, and analyzed the mixing processes of water vapor with different sources. Wang and Yamanaka (2014) developed a new method based on a two-source model for the growing season, and demonstrated the behavior of vegetation affected by physiological responses. Since then, numerical models of isotopic tracers have been developed (Wang et al., 2015), from

which the Iso-SPAC model has evolved, with a steady-state assumption for the transpiration flux successfully reproducing seasonal variations of all the components of the surface-energy balance.

In recent decades, more attention has been paid to climate change, with the need for more comprehensive and diverse data sources. The advantage of a single-site database is the ability to provide refined, high-quality data, since a large-scale database can easily overlook specific procedures occurring within certain events involving both water and energy transport.

The purpose of this paper is to present the available data and the processes involved in their collection for providing high-quality observations over multiple frequency scales for assisting climate studies and energy-balance estimations in multiple ways, such as: 1) providing a complete dataset for water and temperature observations for modeling studies or theoretical investigations; 2) helping to compare forecasted and estimated values of the climate trend; 3) showing a unique dataset representing the meteorological and hydrological variations located in the north-east of Japan; 4) providing a valuable and

unique database for biological and ecological studies as a long-term, continually well-maintained grassland.




## 2 Data description

### 2.1 Observational Site, Data and Instrumentation

The data were collected from an observational site located at the department of Environmental Dynamics and Prediction at the Center for Research in Isotopes (36.0°06'35''N, 140°06'00''E) in the grounds of the University of Tsukuba (Fig. 1a).

This region has a semi-humid marine climate with a long-term-average annual precipitation of 1200–1600 mm (Hamada et al., 1998). The observational site consists of a circular field of diameter 160 m covered by grass at an altitude of 27 m above sea level (a.s.l.), a meteorological observation tower of 30-m height equipped with sensors at heights of 1.6 m, 10 m and 29.5 m, and underground sensors at depths of 0.02 m, 0.1 m, 0.5 m, and 1 m, which monitor the soil temperature and heat flux (Fig. 1b and Tab. 1). All observed time series have been aggregated at frequencies of 30 min, 1 h and 24 h since 1981, and at

a frequency of 10 s since 2003. Being a continuous, long-term observational site, the maintenance and modernization of instrumentation has taken place to ensure high-quality observations, with the date when each instrument was introduced found in Table 1.

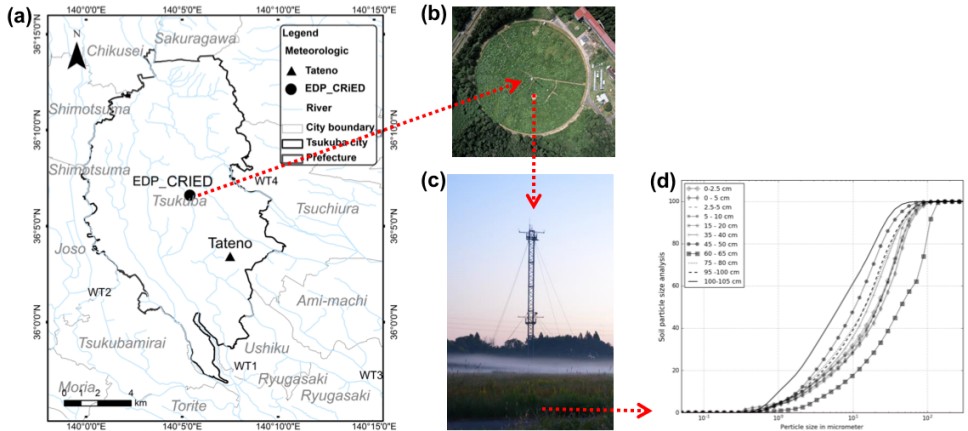

**Figure 1.** EDP observational site: (a) location; (b) projected view of the EDP grassland; (c) the observation tower; (d) results of
the soil content analysis.

The observational site was artificially filled with loam and volcanic ash soil in the top few meters. Each soil layer consists of brown to dark sand, clay and loam in different ratios. A soil-particle-size analysis was carried out using a Shimadsu SALD-3100 particle analyzer, with the results shown in Fig. 1d. During the growing season, the lowest soil layer

encounters groundwater at a depth of around 2 m. The observational site is covered by multiple species of vegetation, with a similar percentage of species each year since 1977 (Imasu, *et al.*, 2002) providing consistent meteorological and hydrological research conditions throughout this period.





The vegetation is naturally grown C3 and C4 vegetation, such as *imperata cylindrical, audropogon virginicus, miscanthus sinensis* as C4, and *solidago altissima, artemisia princeps, lespedeza cuneate, lespedeza pilosa, equisetum arvense, festuca arundinacea, potentilla freyniana, lysimachia clethroides* as C3. The similarity of grass species and their depth, as well as their leaf-area index (LAI), were also explicitly confirmed each year by two different surveys. Tanaka and

Oikawa (1998) conducted a survey of the seasonal dynamics of the LAI for both C3/C4 vegetation types, with results showing that the percentage of the species and the height of the vegetation not changing dramatically since 1977. Another survey was carried out two years later to directly measure the LAI and height from 2000 to 2002, with similar results found as before, such as the value of the LAI, which ranged from less than 0.1 in winter to nearly 5 during September, while the height of the grass is about 1 cm during winter, and around 1.8 m in September and October (Imasu, *et al.*, 2002).

Maintenance for the grass cover includes mowing twice per year in summer and winter since 2006. The dead plants and mowed litter are scattered randomly on the ground to ensure the coverage of the ground with a thick litter layer for most of the year.

**Table 1. Instrumentation information of the EDP observational system.**

| Item | Elevation (m) | Period | Instrument | Maker (Model) |
|---|---|---|---|---|
| Shortwave Radiation | 1.5 | 2015.12~ | Four component radiation balance meter | Hukseflux(CHF-NR01) |
|  | 1.5 | 1981.8~* | Thermocouple type total solar pyrometer | EKO(MS-402F) |
|  |  |  |  | EKO(MS-43F) |
|  |  |  |  | EKO(A75510(MS43F)) |
|  |  |  |  | EKO(A84517) |
| Net Radiation | 1.5 | 2014.3~ | Four component radiation balance meter | Hukseflux(CHF-NR01) |
|  |  |  | Draft-type thermocouple type radiation budget meter | EKO(MF-11, CN-11) |
|  |  | 1981.8~ |  |  |
| Air Temperature** | 1.6, 12.3, 29.5 | 1981.8~ | Ventilated platinum resistance thermometer | Vaisala(CVS-HMP155D) |
|  |  |  | Humidity temperature probe | Vaisala (CVS-HMP45D) |
|  |  |  |  | Nakaasa(E-731) |
| Dew Point Temperature (Relative Humidity) | 1.6, 12.3, 29.5 | 1981.8~2006 | 2006~Calculated from Relative humidity | Vaisala(CVS-HMP) |
|  |  |  | Lithium chloride dew point thermometer | Nakaasa(E-771) |
| Sensible Heat Flux | 1.6 | 1981.8~ | Ultrasonic wind speed thermometer | KAIJO SONIC(DA650, TA-61A) |
|  | 12.3 | 1981.8~1987.8 |  | KAIJO(DA600) |
|  | 29.5NW | 1981.8~ |  | Kaijyo Denki |
|  | 29.5SE | 1981.8~2015.3.26 2015.7.30~ |  |  |
| Soil Temperature | -0.02, -0.1, -0.5, -1.0 | 1981.8~ | Waterproof type platinum resistance thermometer | C-PTG-10 Nakaasa(E-751) |
| Soil Heat Flux | -0.02 | 1981.8~ | Thermocouple type ground heat flow plate |  |
|  |  |  |  | EKO(CN-81) |
|  | -0.1 | 2013.5.21~ | Heat flow plate | Hukseflux(HFP01-10) |
| Wind Speed | 1.6 | 1981.8~ | Ultrasonic wind speed thermometer | KAIJO SONIC(DA650, TA-61A) |
|  | 12.3 | 1981.8~1987.8 |  | KAIJO SONIC(DA600) |



| | 29.5NW | 1981.8~ | | Kaijyo Denki |
| | 29.5SE | 1981.8~2015.3.26 2015.7.30~ | | |
| | 30.35 | 2014.2.28~ | Propeller type wind direction anemometer | YOUNG(CYG-5103LM) |
| Precipitation | 0.3 | 1981.8~ | Tipping falls Separated type self-recorded rain gauge | Yokokawa(WB0013-05) |
| | | | | Yokokawa(B-011-00) |
| Atmospheric Pressure | 1.5 | 1981.8~ | Barometer | Vaisala(PTB210) |
| | | | | Nakaasa(F-401) |

\* Frequency for all observation periods: 0.5, 1, and 24 hours. A frequency of 10 seconds was used starting in 2003.
\*\* Air temperature observations included average, maximum and minimum values.

Most observations we present were observed directly, except for the dew point temperature $T_d$, which, according to the
historical record, has been collected by different instrumentation. While the initial instrumentation for the dew point
temperature was a Lithium chloride dew point thermometer, since 15 December 2006, this observation has been changed to a
hygrometer (CVS-HMMP45D, Climatec), with the value of the dew point temperature calculated from the observed specific
humidity by (Watarai and Yamanaka, 2007)

$$T_d = \frac{b \times \log_{10}\left(\frac{e}{6.11}\right)}{a - \log_{10}\left(\frac{e}{6.11}\right)},\tag{1}$$

$$e = e_{sat} \times \frac{RH}{100},\tag{2}$$

where $a = 7.5$, $b = 237.3$, $RH$ is the relative humidity expressed as a percentage, and the saturated vapor pressure $e_{sat}$ is
obtained from

$$e_{sat} = 6.11 \times 10^{\left(\frac{aT}{b+T}\right)},\tag{3}$$

where the temperature $T$ is given in °C. Data collected from supersonic anemometer–thermometers are used to obtain the
heat and momentum fluxes by the eddy-correlation method. These observational data are open to the public and are freely
available through our website (see "http://www.ied.tsukuba.ac.jp/en/edps/database-doi/", which is renewed every minute,
and TERC is the former name of the EDP department). Observations are recorded every 30 min (since 1 May 2003), which
are then converted into daily averaged data (Asanuma et al., 2004) provided at least 20 readings are recorded; otherwise, the
day is marked as a missing record, with days with more than 20 and less than 24 readings marked as incomplete (Ohba and
Yamanaka, 2007). In addition to the missing data, the dates of equipment maintenance, as well as all construction and
mowing          information,          are          recorded          in          the          maintenance          log          accessible          at
http://www.ied.tsukuba.ac.jp/yosoku/kansoku/hojyo_log/.



## 2.2 Estimation of downward longwave radiation

Using a method developed by Kondo and Xu (1997), the downward longwave radiation flux is estimated from the routine meteorological observations, including the solar radiation, air temperature, relative humidity, air pressure, wind speed, and precipitation (see Table 1 for instrumentation information), with the temporal resolution of the calculation chosen

based on both the observed forcing data and the estimated downward longwave radiation. The results of the estimation have been validated against the downward longwave radiation as directly observed by the EDP department between 2002 and 2007. For assessing the model performance, the coefficient of determination ($R^2$), Nash–Sutcliffe efficiency ($NSE$), and the percent bias ($PBIAS$) are considered.

### 2.2.1 Estimation of downward longwave radiation

The downward longwave radiation at the ground is estimated following Kondo et al. (1994) as

$$L^{\downarrow} = \sigma T^4 \left[ 1 - \left( 1 - \frac{L_f^{\downarrow}}{\sigma T^4} \right) C \right] \tag{4}$$

$$L_f^{\downarrow} = (0.74 + 0.19x + 0.07x^2)\sigma T^4 \tag{5}$$

$$x \equiv \log_{10} w^* \tag{6}$$

where $L^{\downarrow}$ is the downward longwave radiation for a cloudy day, $L_f^{\downarrow}$ is the downward longwave radiation for a fine day, $T$ (K)

is the 1-h air temperature, $\sigma$ =5.670×$10^{-8}$ W m$^{-2}$ K$^{-4}$ is the Stefan–Boltzmann constant, and $w^*$ is the effective precipitable water expressed as

$$w^* = \frac{1}{g} \int_0^{P_s} q \frac{p}{p_0} dp \tag{7}$$

Here, $p_0 = 1013$ hPa is the standard atmospheric pressure, and $p_s$ is the surface pressure (hPa). The precipitable water (cm) is estimated as

$$\log_{10} w \approx \log_{10} w^* + 0.10 \tag{8}$$

and $C$ is the coefficient expressing the effect of clouds by

$$\begin{aligned} C &= 0.03B^3 - 0.03B^2 + 1.25B - 0.04 & (B \geq 0.0323) \\ &= 0 & (B < 0.0323), \end{aligned} \tag{9}$$

where $B \equiv S_{obs}^{\downarrow} / S_{top}^{\downarrow}$. Here, $S_{obs}^{\downarrow}$ is the observed solar radiation flux, and $S_{top}^{\downarrow}$ is the mean downward solar radiation at the

top of the atmosphere,





$$S_{top}^{\downarrow} = \frac{S_{00}}{\pi} d(\zeta \sin \phi \sin \delta + \cos \phi \cos \delta \sin \zeta)$$

(10)

where

$$\zeta = \cos^{-1}\left(\frac{\sin \alpha - \sin \phi \sin \delta}{\cos \phi \cos \delta}\right)$$

(11)

$$d = 1.00011 + 0.034221 \cos \eta + 0.00128 \sin \eta$$
$$+0.000719 \cos 2\eta + 0.000077 \sin 2\eta$$

(12)

$$\delta = \sin^{-1}[0.398 \times \sin(4.871 + \eta + 0.033 \sin \eta)]$$

(13)

$$\eta = \frac{2\pi}{365} Day$$

(14)

$S_{00}$ is the solar constant, $\zeta$ is the half-day angle, $\phi$ is the latitude, $\delta$ is the solar declination, and *Day* is the total number of days from 1 January to the day of observation. According to Kondo et al. (1994), the estimated downward longwave radiation is accurate to within $\pm 10$ W m$^{-2}$, with differences resulting from the effects of snow in winter.

### 2.2.2 Model evaluation

Evaluating the agreement between observed and simulated data is a basic requirement for the data quality control (Moriasi et al., 2007, Singh et al., 2005) for which the following indices are chosen: the coefficient of determination ($R^2$) for describing the ratio of variance in the observations; the Nash–Sutcliffe efficiency (*NSE*) for indicating the error in the units of the constituent between observed and simulated data; and the percentage bias (*PBIAS*) for measuring the average tendency of the simulated data to be larger or smaller than the corresponding observed data (Gupta et al., 1999). Here,

$$NSE = 1 - \left[\frac{\sum_{i=1}^{n}\left(Y_i^{obs} - Y_i^{sim}\right)^2}{\sum_{i=1}^{n}\left(Y_i^{obs} - Y^{mean}\right)^2}\right]$$

(15)

$$PBIAS = \left[\frac{\sum_{i=1}^{n}\left(Y_i^{obs} - Y_i^{sim}\right)*(100)}{\sum_{i=1}^{n}\left(Y_i^{obs}\right)}\right]$$

(16)

where *n* is the total number of observations, $Y_i^{obs}$ is the observed value for the $i^{th}$ step, $Y_i^{sim}$ is the simulated value, $Y^{mean}$ is mean value for all the observed value (Moriasi et al., 2007). Here, a *NSE* value of unity represents the optimal value, while the optimal value is zero for the *PBIAS*, with low-magnitude values indicating an accurate simulation. These modeling indices are evaluated for checking the accuracy of the time series of the estimated downward longwave radiation.





## 3 Data quality control

The data quality has been checked for the entire observation period, including errors deleted resulting from maintenance activities or abnormal system behavior, and marked as missing values in the EDP database, with Table 2 showing the data availability within the database after error deletion. To preserve only original observational data from the EDP site, we do

not conduct any gap filling.

**Table 2.** Parameter information within the EDP database. Location A, B and C are located within the observation site of the EDP department as shown in Fig. 1.

| Item | Unit | Height (m) | Period | Data coverage (%) | Average | Maximum | Minimum |
|---|---|---|---|---|---|---|---|
| Shortwave downward radiation* | W/m² | 1.5 | 1981~* | 94.64 | 146.63 | 356.50 | 3.50 |
| Net radiation | W/m² | 1.5 | 1981~ | 93.84 | 67.39 | 237.20 | -15.00 |
| Longwave downward radiation | W/m² | 1.5 | 2002-2007** | 84.75 | 346.51 | 450.75 | 227.37 |
| Air temperature_Layer1_average* | °C | 1.6 | 1981~ | 95.58 | 13.93 | 31.10 | -5.30 |
| Air temperature_Layer1_maximum* | °C | 1.6 | 1981~ | 98.26 | 19.06 | 37.40 | -0.60 |
| Air temperature_Layer1_minimum* | °C | 1.6 | 1981~ | 97.30 | 9.03 | 26.70 | -11.30 |
| Air temperature_Layer2 | °C | 12.5 | 1981~ | 97.79 | 14.23 | 31.10 | -3.80 |
| Air temperature_Layer3 | °C | 29.5 | 1981~ | 96.11 | 14.51 | 31.20 | -3.00 |
| Dew point temperature_Layer1 | °C | 1.6 | 1981~ | 95.51 | 8.86 | 26.30 | -14.70 |
| Dew point temperature_Layer2 | °C | 12.5 | 1981~ | 89.46 | 8.68 | 27.50 | -19.50 |
| Dew point temperature_Layer3 | °C | 29.5 | 1981~ | 89.60 | 8.53 | 27.30 | -15.80 |
| Soil temperature_Layer1 | °C | -0.02 | 1981~ | 97.45 | 14.79 | 32.30 | -1.50 |
| Soil temperature_Layer2 | °C | -0.10 | 1981~ | 97.73 | 14.72 | 30.30 | 0.60 |
| Soil temperature_Layer3 | °C | -0.50 | 1981~ | 92.50 | 14.95 | 25.90 | 3.10 |
| Soil temperature_Layer4 | °C | -1.00 | 1981~ | 97.63 | 14.78 | 25.20 | 5.00 |
| Sensible heat flux | °Cm/s | 1.6 | 1981~ | 90.91 | 0.01 | 0.07 | -0.03 |
| Soil heat flux | W/m² | -0.02 | 1981~ | 85.13 | -0.34 | 20.00 | -20.00 |
| Wind speed* | m/s | 1.6 | 1981~ | 99.44 | 0.97 | 3.80 | 0.16 |
| Relative humidity* | % | 1.6 | 2003~ | 97.17** | 75.25 | 100.10 | 28.11 |
| Precipitation* | mm/hour | 0.5 | 1981~ | 96.51 | 3.20 | 181.50 | 0.00 |
| Atmosphere pressure | hPa | 1.6 | 1983~ | 90.67 | 1010.41 | 1033.60 | 977.30 |

\* Data items compared with data from the Japan Meteorological Agency (JMA_Tateno)

\*\* The data coverage for relative humidity was estimated starting in 2003.

\*\*\* The observation of longwave downward radiation was between 2002 and 2007. Data coverage, average, maximum and minimum were determined using estimated longwave radiation based on other observation items. The estimated longwave radiation data comprise dates since 1983.

## 3.1 Comparison with the Japan Meteorological Agency

After the deletion of error values, the reliability of the observations from the EDP database is assessed with respect to the reference database from the nearest local meteorological agency in Tateno, which belongs to the Japan Meteorological Agency. The Tateno observations are at a height of 2 m above ground level, with only a few items available for data-quality checking, including the shortwave radiation, air temperature (average, maximum and minimum), wind speed, relative



humidity and precipitation, at a daily resolution between 1981 and 2017 (see Table 2, Fig. 2). For the EDP database, observations from the 1.6-m height are compared with the Tateno data, with the data at other heights compared with the adjacent measurement heights. Both of these two datasets are compared in Fig. 2 along with the corresponding values of $R^2$.

As the value of $R^2 > 0.99$ for the average, maximum and minimum values of the air temperature in correlations between the Tateno and EDP databases, a high similarity exists between the two closely located stations separated by < 10 km, and an elevation difference of 1.8 m (the Tateno and EPD sites are 25.2 m and 27 m a.s.l., respectively). However, slight differences between these two databases were found, which is caused by the different land surface, such as the presence of different vegetation and artificial structures. Because of the diverse land-surface cover and varied moisture distribution, the value of the relative humidity $RH$ is slightly difference, but still well correlated. While shortwave radiation is mainly governed by the

solar radiation, the absorption and reflection caused by the presence of clouds and other constituents in the atmosphere result in plausible differences between the EDP and Tateno databases. Overall, most of the variables are highly correlated, except the wind speed and precipitation.

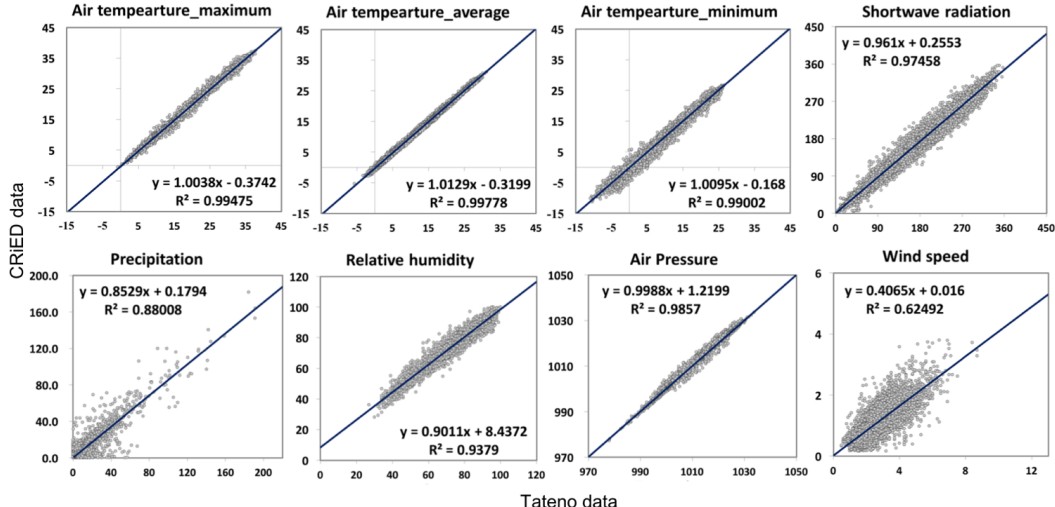

**Figure 2. Comparisons of the EDP data with the Tateno data from the Japan Meteorological Agency from 1981 to 2017.**

**Further parameter information is found in Table 2.**

According to the regression analysis, a lower correlation is found for the precipitation and wind speed between these two stations (Fig. 2), because of the highly local characteristics of these two variables. In the case of the wind speed, the sensor of the EDP site located at a height of 1.6 m is surrounded by grass during the growing season and prior to mowing

when the height of the grass reaches 1.8 m, which generates enhanced local turbulence, resulting in a reduced wind speed.




The precipitation amounts are significantly affected by the local conditions depending on the humidity, temperature, cloud distribution and wind speed (Shuttleworth, 2011). In general, after error deletion, the EDP database compares reasonably to the Tateno database.

### 3.2 Downward longwave radiation

The downward longwave radiation was calculated by using the method introduced in Sect. 2.2.1 as estimated from routine meteorological observations from 1983 to 2017, and compared with a five-year dataset collected by the EDP department from 2002 to 2006 (Fig. 3), giving $R^2 = 0.974$, $NSE = 0.947$ and $PBIAS = 1.486$, and indicating good agreement of the estimated values with the observed values according to Moriasi et al. (2007). Although the estimated values are slightly higher than the observed values, this slight bias of estimation is consistent with the statement by Kondo et al. (1994)

regarding the effect of winter resulting from the presence of snow. However, most of the estimated values correspond well with the observed values.

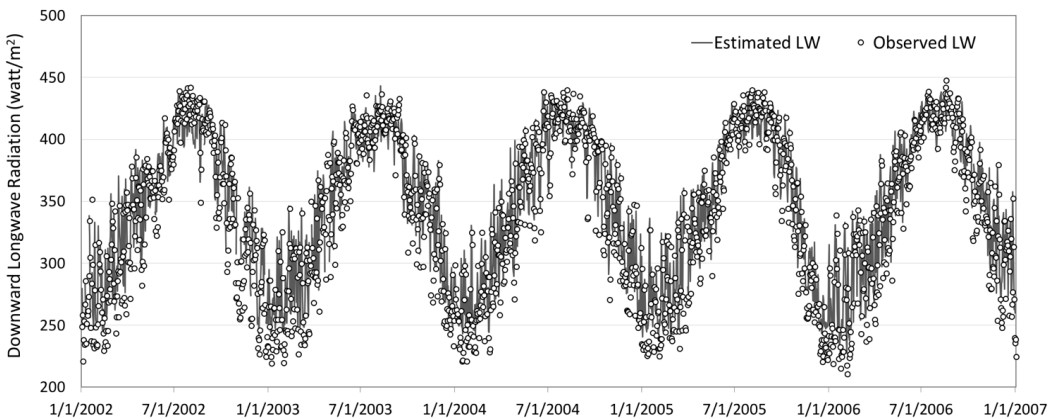

**Figure 3. Comparison of estimated and observed downward longwave radiation at the EDP site from 2002 to 2006.**

### 3.3 Statistical analysis

Plotted in Fig. 4 are daily observed values of the air and soil temperatures for all layers, the precipitation, air pressure, humidity, longwave radiation, solar radiation, net radiation ($Rn$), the soil heat flux ($G$) and sensible heat flux ($H$) after data quality control. For each soil layer, the temperature varies between the deep and shallow layers, with the long-term-average values varying from 14.7°C to 15°C, the maximum values from 25.2 °C to 32.3 °C, and the minimum values from −1.5 °C to

5°C. A regular annual variation is found for the four soil layers, with the response of the deeper layers slower than the shallow layers. The larger amplitude variation is evident in the shallowest soil layer, with the amplitude gradually decreasing



for deeper soil layers. Some extremely high values occurred in 1990, 1991 and 2005, as well as extremely low values from 1983 to 1985, and 2011 to 2013. In contrast to the temperature of the different soil layers, the three air-temperature measurements show similar patterns of variation, with the 37-year average values for the heights of 1.6 m, 12.5 m and 29.5 m corresponding to 13.9°C, 14.2°C and 14.5°C, respectively. The maximum value is 31.1°C for all three layers, but the

minimum values show obvious differences of −5.3°C at 1.6 m, −3.8°C at 12.9 m, and −3.0°C at 29.5 m. For the dew point temperature, the three layers show very similar ranges of observation values. From 1.6 m to 29.5 m, the long-term average varies from 8.5°C to 8.9°C, the maximum value from 26.3°C to 27.9°C, and the minimum value from −14.7°C to −19.5°C. The minimum values for the air and dew point temperature at 1.6 m are more different than the other layers, since the 1.6-m measurement height is more easily affected by any dramatic temperature or moisture exchanges with the ground.

The 37-year average value of precipitation is 3122.1 mm per year, and the maximum precipitation event occurred in 1986 with an amount of 181.5 mm per day. The relative humidity, which takes an average value of 75.2% for the 37-year period. The long-term-average value of the air pressure is 1010.4 hPa.

The long-term-average value of the net radiation is 66.9 W m$^{-2}$, while the maximum value is 237.2 W m$^{-2}$, and the minimum is −15 W m$^{-2}$. The inter-annual variation indicates an increasing tendency for both the summer and winter from

1981 to 2004, but which has reduced since 2005. The average value of the shortwave radiation is about 146.6 W m$^{-2}$, while the maximum value is 356.5 W m$^{-2}$, and the minimum is −3.5 W m$^{-2}$. The average value of the longwave radiation is 346.5 W m$^{-2}$, while the maximum value is 450.8 W m$^{-2}$, and the minimum is 227.4 W m$^{-2}$. The soil heat flux and sensible heat flux show less regular variations compared with the values of the temperatures presented above, because of the difficulties in the accurate monitoring of the soil heat flux resulting from the high spatial variation of the soil properties. Furthermore, the soil

heat flux is easily influenced by the soil temperature, air temperature and heat capacity, as well as the thermal peculiarities of heterogeneous soil layers. However, since the observation of these thermal components is rare, they are highly valuable for hydro–meteorological investigations. Further analysis of the database is highly encouraged for specific research topics.







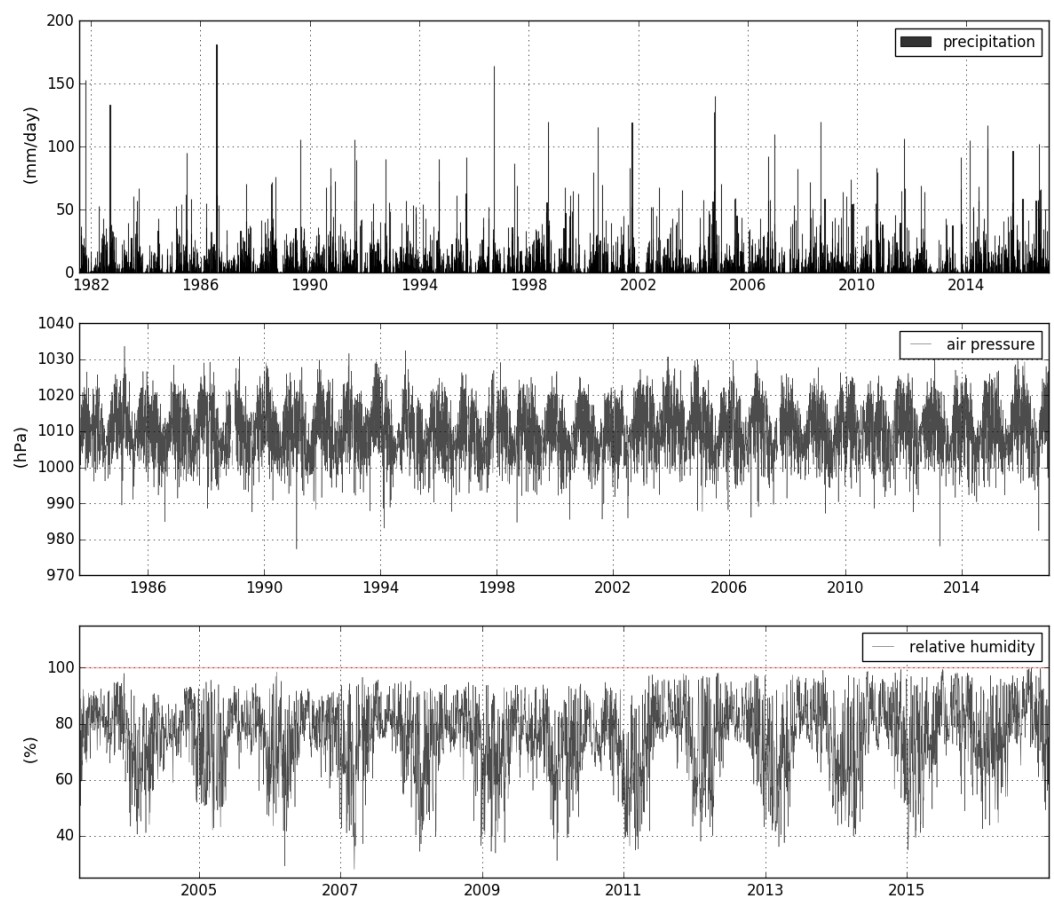



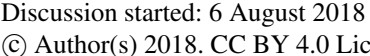

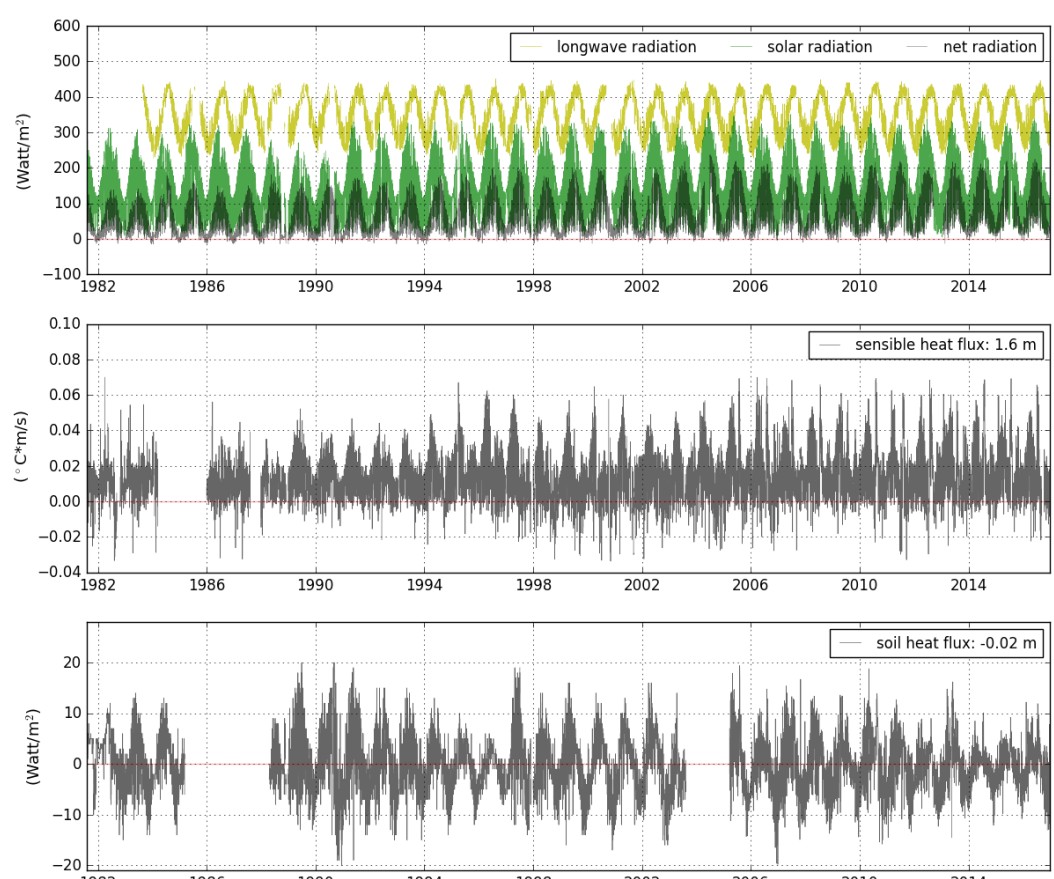

**Figure 4. Observed soil temperature, dew point temperature, net radiation, soil heat flux and sensible heat flux at the EDP site from 1981 to 2017.**

## 4 Results

5    The complete long-term hydro–meteorological dataset at our grassland site represents a valuable resource for climate studies. For example, Duan et al. (2015) demonstrated the uniqueness of this area, which indicates changes in the precipitation amounts in Japan are in the opposite direction to the extremes seen in the surrounding countries. Moreover, the database is capable of providing data for assessing both the energy and water budgets. Here, the main tendencies in the last



37 years are summarized in Fig. 5, where the annual values represent the years with a data availability > 90% (328 days for one year), because gap filling has not been performed.

Based on the annually averaged values, most variables show a positive trend, except the soil temperature, soil heat flux, wind speed, and precipitation, which show a slightly negative trend. First, the shortwave, longwave and net radiation

increased between 1981 and 2017 (Fig. 5, a-c). The shortwave radiation was compared with data from the Tateno database above, from which a similar positive tendency was found. That the estimated longwave radiation has a positive trend is consistent with the air-temperature trend with reference to the Stefan–Boltzmann law (Shuttleworth, 2011). Since the shortwave radiation has increased, it is reasonable to expect the net radiation to have increased as well, because the balance of net radiation is governed by the balance of the shortwave and longwave radiation. An observed positive air-temperature

trend is evident since 1981 (Fig. 5, j), which is consistent with the global trend related to climate change and an increasing temperature. The average, maximum and minimum values of the air temperature all show a positive trend with an increase of 0.04°C per year. Here, the minimum value increased more slowly than the other two statistics, because the minimum air temperature is highly affected by urbanization (Kondo et al., 1994), where the main construction period of Tsukuba occurred within the 1980s, and slowing down thereafter, so that the minimum air temperature shows a relatively smaller positive trend

compared with the maximum and average values. The dew point temperature mainly followed a similar positive trend as the air temperature, with similarly increasing values at all three heights, including the first height, which is often surrounded by grass (see Fig. 5, k, Fig. 1). The variation of the surface soil layer (−0.02 m) shows a higher amplitude than the lower soil layers, which are more exposed to the energy exchange occurring in the lower atmosphere, and since the EDP site is covered by a bushy grass, the energy transport received has a strong influence. Therefore, the temperature of the soil layer does not

show any simple deviation from the annual average values (Fig. 5, l). The explanation of the negative tendency of the soil heat flux (Fig. 5, e) relates to the energy balance as pointed out by Brutsaert (1981), where $R_n = L_eE + H + G$, $L_e$ is the latent heat of vaporization, and $L_p$ is the thermal conversion factor for the fixation of carbon dioxide, so that the variation of the soil heat flux ($G = R_n\text{-}L_eE\text{-}H$) is directly related to the variation of the net radiation, and the latent and sensible heat fluxes. The tendency of evapotranspiration is possibly positive since the temperature, which is the dominant factor, shows a

significant positive trend. Furthermore, although there is a slightly positive trend of net radiation, as well as a negative trend of the sensible heat flux, the soil heat flux shows a negative trend. The corresponding increase of the potential evapotranspiration is very possibly caused by the air temperature rising, which is in accordance with global climate change. As the temperature is the most important parameter for evapotranspiration, the increase in temperature probably leads to an increase of the evapotranspiration, meaning the decrease of the soil heat flux is reasonable, although no data exist for the

observations of the latent heat flux. The sensible heat flux and the relative humidity show a positive trend (Fig. 5, d, i). The main variation of the trend in the precipitation is relatively moderate, because of the more extreme low and high values recorded in recent decades, which is consistent with the more extreme precipitation observed globally (Donat et al., 2016,



Fig. 5, h). Compared with precipitation, a similar trend may be found in the relative humidity, although the analysis period is relatively short, beginning 2004 (Fig. 5, i). For the wind speed (Fig. 5, f), there is negative tendency shown for the long-term observations, which is easily affected by the variation of the surrounding land cover. During these decades, two minor decreases in the signal are found between 1993 and 2002, which is also found in the value of the air pressure (Fig. 5, g).

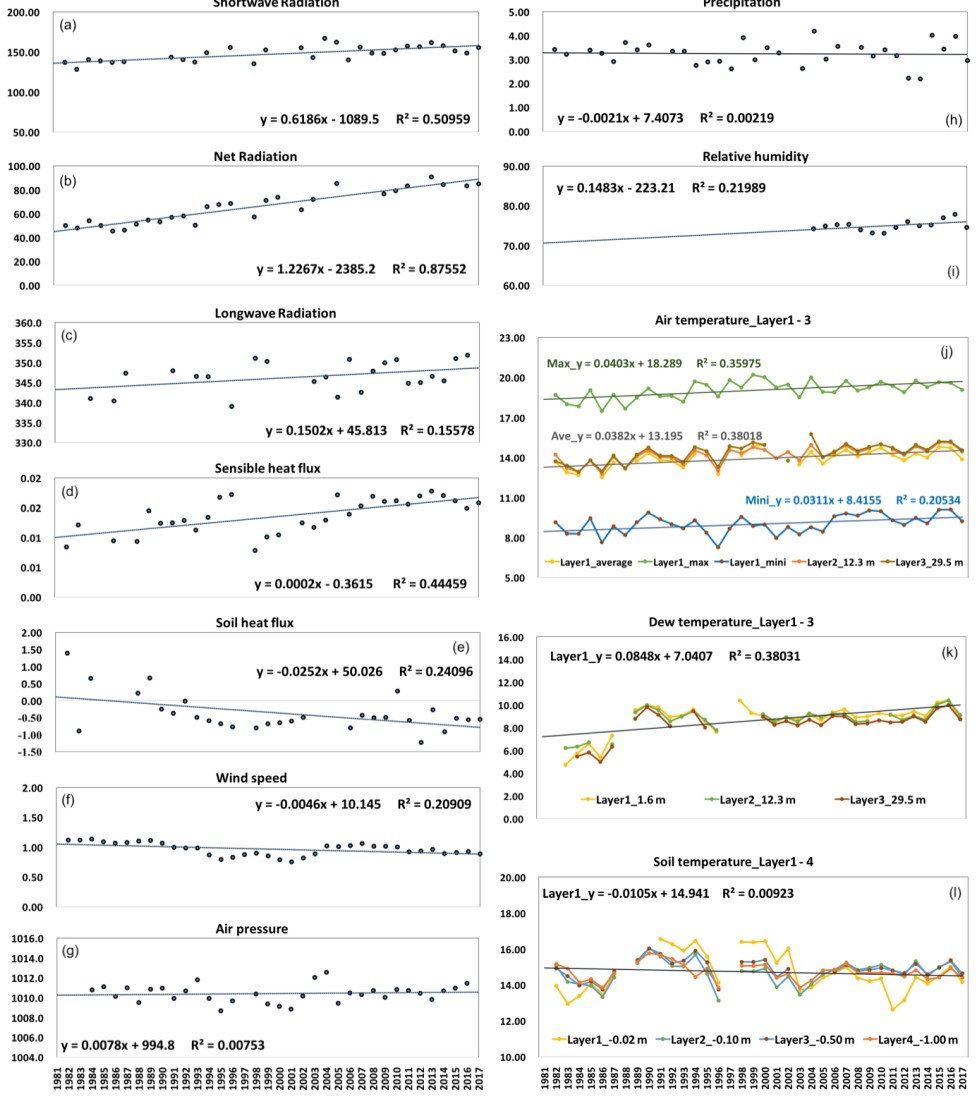



**Figure 5. Annual average values of the observational database showing the (a) shortwave radiation (W m⁻²), (b) net radiation (W m⁻²), (c) longwave radiation (W m⁻²), (d) sensible heat flux (°C m s⁻¹), (e) soil heat flux (W m⁻²), (f) wind speed (m s⁻¹), (g) air pressure (hPa), (h) precipitation (mm per day), (i) relative humidity, (j) average, maximum and minimum air temperature (°C) at a height of 1.6 m, with the average values at the heights of 12.3 m and 29.5 m, (k) dew point temperature (°C) at heights of 1.6 m, 12.3 m and 29.5 m, and (l) soil temperature (°C) at the depths of 0.02 m, 0.1 m, 0.5 m and 1 m.**

## 5 Data availability

The database described here has a Digital Object Identifier (doi:**10.24575/0001.198108**), and is freely available at the home page of the EDP data center (http://doi.org/10.24575/0001.198108). The data must be fully referenced for every use as introduced at http://www.ied.tsukuba.ac.jp/en/edps/database-doi/. Supplemental materials may be found by checking the observational data web site at http://www.ied.tsukuba.ac.jp/yosoku/kansoku/. Maintenance information is found from the database log at http://www.ied.tsukuba.ac.jp/yosoku/kansoku/hojyo_log/.

## 6 Conclusions

A high-quality database covering 37 years is presented from the north-east of Japan, encompassing four observational frequencies, three measurement levels above ground, and four layers below the ground. The daily and annual average values were presented, including the daily values of shortwave radiation, average, maximum and minimum air temperatures, wind speed, relative humidity and precipitation, which were compared with data from the local meteorological agency at Tateno for validation of the data quality. Highly valuable data are also presented, such as the net radiation, soil temperature at four depths, the air and dew point temperatures at three heights, the net radiation, soil heat flux, and sensible heat flux, with especially the soil heat flux and sensible heat flux relatively rarely observed or available. Furthermore, a time series of longwave radiation has been generated for this site based on the reliable observations.

The annual average values were analyzed for the years with data availability > 90%. The annual average values show a positive tendency for the shortwave radiation, net radiation, longwave radiation, sensible heat flux, air pressure, relative humidity, the air and dew point temperatures at the three heights, and a negative trend of the soil heat flux, wind speed, precipitation and soil temperature. These trends provide for an important database and evidence for understanding the variations occurring within the study area. At the same time, the specific characteristics of the database may be found for the grassland with respect to the values of the wind speed and precipitation based on the comparisons between the daily values from the EDP database and Tateno, as well as the regression of annual average values. As the in situ observational site and database have contributed to many previous studies, we shall continue to maintain it, and we wish to make the data available for further research and analysis.



## Author Contribution

Jun Asanuma created the observation system, data collection system and controls the measurement. Wenchao Ma prepared the database and wrote this manuscript. Jianqing Xu provided the theoretical support of downward longwave radiation. Wenchao Ma and Jianqing Xu estimated the downward longwave radiation. Jun Asanuma and Yuichi Onda designed this work. All authors provided critical feedback and helped shape the research, analysis and manuscript.

## Acknowledgements

This study was supported by "Interdisciplinary Project on Environmental Transfer of Radionuclides". The authors would like to acknowledge Prof. Tomohiro Sekiguchi and Dr. Kentaro Aida for their assistance in the system maintenance. We thank Hideo Iijima, Shiraishi Izumi and Naomi Nakajima for providing digital and paper based bulletins for this review.

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
