# Peer review of "A database of water and heat observations over grassland in the north-east of Japan"

_Earth System Science Data, 2018_

## Referee Comment (RC1) · Anonymous Referee #1 · 22 Aug 2018

* * *
General Comment

The database comprising 37 years of data from the EDP observational site is impressive. Various parameters are recorded in different soil depth and instruments heights, and various temporal resolutions are also offered. The data page is clear (in the English version) and the download easy. The description should maybe also contain a little paragraph on instrument calibration and maintenance (e.g. cleaning, which is especially relevant for radiation sensors).

I recommend the publication of this important database, but have a few remarks as well.

[Figure]

———————————————————-

Specific comments

Datafiles ==> The headers of the files have to be copied and inserted into the data files, which themselves do not contain headers. As this is probably done for automatic ingest I am fine with it. However, the last data point in the hourly files is always hour 24:00. E.g. hour 24:00 of day 1, whereas I would have given this datapoint the timestamp hour 00:00 of day 2. Is that Japanese standard maybe? Other than that, the files are in good shape and easily processable.

P4 Line 17 ff ==> I am not sure about the representativeness of this study area for plant/soil studies. If the site was artificially filled this is not representative of the area nor natural conditions. Also, the area around the site looks more of an agricultural/urban/foresty site – how representative is this site of the area? Please comment on that.

P6 Line 17 ff ==> You talk about the averaging process here. I would like to hear more of how you averaged the data for hourly or daily averages, especially for data with large daily courses (shortwave radiation), this is pretty important if serious data gaps exist (or e.g. only nighttime values are available). How is a record marked as icomplete? I did not see any marks in the files I looked at.

P6 Line 20 ==> the maintenance log is a good thing, but I only saw the Japanese version – is there an English version available?

P7 Chapter 2.2 ==> I did not find the downward longwave data, can you specify where it is included? See also Table 2, where it seems that longwave data is included in the 2002-2007 files, but I checked 2004 and there is no longwave included

P9 Chapter 3 ==> maybe you should also talk about the different available versions here. What is the difference between version 1.0, 1.1 and 2 etc.

P11 Line 16 ==> you talk about daily observed values – please clarify here, that it is

daily averages (e.g. temperature) and daily sums (e.g. precipitation?) respectively.

P12 Line 10 ==> you state here that the average annual precipitation is 3122 mm/a, and in Figure 5 that seems to be true. BUT on page 4 you say that the region has a long-term annual average precipitation of 1200-1600 mm, and in the 2004 file e.g. the sum of all hourly values for that year amounts to around 1520 mm. So the 3122 mm average, and all values in Fig. 5, seem way too high.

P15 Description of Figure 4 ==> Why do you mention soil and dew point temperature, but not air temperature, which is also included in Figure 4? Also other parameters are not mentioned here, why?

————————————

Technical corrections

P1 Line 13 ==> If you write "depth", normally the numbers are not negative. I would remove the "-" [minus] signs when you write "at depths of -0.02 m, ..."

P4 Line 4 ==> in the lat/long the 36.0° can be reduced to 36° (remove ".0")

P4 Figure 1 ==> Please enhance the size of the font a bit, as the X and Y axis is not easily readable

P5 Line 1-3 ==> Please use capital first letter for the genus, e.g. Imperata cylindrica (no l in cylindrica!), Andropogon virginicus (Andropogon, not Audropogon!) etc.; Lespedeza cuneata is with a at the end, not e (probably all automatic spell check errors)

P5 Table 1 ==> The date format is a little weird here. I would maybe use ISO format (2015-12 or 1981-08 etc.)

P18 Line 3 ==> as you give units for all parameter, you should also list "%" for relative humidity

References ==> you list three references as "in review" (Godsey, Kosmos, Makarieva).

All three have been published in the meanwhile, so you should update the references
* * *

---

## Referee Comment (RC2) · Anonymous Referee #1 · 22 Aug 2018

sorry I forgot the following: In the abstract you cite "Konto and Xu (1997)", but that should probably mean "Kondo and Xu (1997)"

---

## Author Comment (AC1) · 10 Sep 2018

"The database comprising 37 years of data from the EDP observational site is impressive. Various parameters are recorded in different soil depth and instruments heights, and various temporal resolutions are also offered. The data page is clear (in the English version) and the download easy. The description should maybe also contain a little paragraph on instrument calibration and maintenance (e.g. cleaning, which is especially relevant for radiation sensors). I recommend the publication of this important database, but have a few remarks as well."

(Response) We thank the reviewer for the comments. This database is concentrated on the efforts of many researchers. Also, the observation site serves to the hydrological,

meteorological and ecological researches for decades. We would like to present this database to serve more research activities. And within this study, we submit the daily data that is carefully checked. Regarding the comments on the "description should maybe also contain a little paragraph on instrument calibration and maintenance", we agree that we should completely describe the instrument calibration and maintenance. About the instrument calibration and maintenance, we have mentioned in Page 6 Line 20-22: "In addition to the missing data, the dates of equipment maintenance, as well as all construction and mowing information, are recorded in the maintenance log accessible at http://www.ied.tsukuba.ac.jp/yosoku/kansoku/hojyo_log/." We would like to translate them into English version and present a thorough description in the next work. "I recommend the publication of this important database, but have a few remarks as well." We thank the reviewer for his/her recommendation!

Specific comments 1. Datafiles ==> The headers of the files have to be copied and inserted into the data files, which themselves do not contain headers. As this is probably done for automatic ingest I am fine with it. However, the last data point in the hourly files is always hour 24:00. E.g. hour 24:00 of day 1, whereas I would have given this data point the timestamp hour 00:00 of day 2. Is that Japanese standard maybe? Other than that, the files are in good shape and easily processable.

(Response) We thank the reviewer for pointing this out. Within the database (CRiED, http://www.ied.tsukuba.ac.jp/yosoku/database-doi/), the headers were not inserted into each data file. While, the headers were written in the "readme.txt" file, which is saved within the same folder for each database version. For the convenience of the users, we will insert the headers into each data file as reviewer suggested. About the timestamp of the database, the hourly data was recorded from 01:00 to 24:00, because of the hourly value represents the value for one hour before. For example, the data on "24:00" is the average or sum (precipitation) value between "23:00" and "24:00", which recorded as the "ended value". In this work, we prepared the daily value, which calculated data from "01:00" to "24:00". Although the writing format of hourly data is different

to the standard format, the daily value is the same. And here, we submit the daily value only.

2. P4 Line 17 ff ==> I am not sure about the representativeness of this study area for plant/soil studies. If the site was artificially filled this is not representative of the area nor natural conditions. Also, the area around the site looks more of an agricultural/urban/forest site – how representative is this site of the area? Please comment on that.

(Response) The representativeness of the artificial soil is an essential point for this observation site. As it is well known, disturbed soil profile is not easily recovered or reproduced artificially. However, in 1977, researchers refilled this observation site by referring to soil profile nearby. And, all of the soil filling in were chosen from the wildly distributed "Kanto loam", which is the highly representative natural product of the Kanto area in Japan. In the other point, soil property and distribution behaves high heterogeneity for all scales: macro-, meso- or microscale even for undistributed soil. Although, the disturbed soil profile does not represent the original soil condition, but it is possible to represent about the Kanto area to some extent. About the area around the site, we checked the land use and land cover data from Ministry of Land, Infrastructure, Transport and Tourism (http://nlftp.mlit.go.jp/ksj-e/index.html). We analyzed percentage of the land use in Tsukuba city using shape data based on the investigation carried out in 2014. The resolution of mesh is 100 meter. Due to the limited resolution of the image, we only present the land use of ArcGIS figure in the supplement rather than put into the manuscript. From the statistical analyzation: 27.3% is farmland, 24.6% is building, 19.4% is paddy field, 16.8% is forest, 6% is others, 2.3% is river, 1.3% is road, 1.1% is golf course, 1.0% is uncultivated land and 0.1% is railway. Basing on the above information, we generally consider the grassland area is about 47.8% and forest is 16.8%, which means the grassland is nearly half of the area of Tsukuba city. That is why, we believe the CRiED grassland observation site is meaningful and representative for this study area.

3. P6 Line 17 ff ==> You talk about the averaging process here. I would like to hear more of how you averaged the data for hourly or daily averages, especially for data with large daily courses (shortwave radiation), this is pretty important if serious data gaps exist (or e.g. only nighttime values are available). How is a record marked as icomplete? I did not see any marks in the files I looked at.

(Response) About the averaging process, we followed the method introduced by Asanuma et al., (2004). The 10 sec data is instantaneous value. For precipitation, the 30 mins, 1 hour and 24 hours data are accumulated values. For the other parameters, the 30 mins, 1 hour data and 24 hours data are average values. For shortwave radiation, the missing/incomplete values only refer to day time when receiving radiation. Also, same as the "question 1", the hourly value, the timestamp value represent the value obtained within 1 hour before. And the same treatment was applied to other time scale frequency. We thank the reviewer for mentioning this database mark. For the datafile we submitted last time, we presented daily data without marking the incomplete data. So, we submit the revised data file in Version 2, the incomplete data are marked with "*", and missing data is marked with "***", which is consistent with description within the manuscript [P6 L17-21]. The modification could be found from the newly submitted supplement data file. Text was added to the context as followed: "(marked with "*" in the supplement data file)" [P6 L18-19] "(marked with "***" in the supplement data file)" [P6 L19-20]

4. P6 Line 20 ==> the maintenance log is a good thing, but I only saw the Japanese version – is there an English version available?

(Response) Thank you for pointing this out. We do not have the English version about the maintenance log yet. The full English version will be prepared soon.

5. P7 Chapter 2.2 ==> I did not find the downward longwave data, can you specify where it is included? See also Table 2, where it seems that longwave data is included in the 2002-2007 files, but I checked 2004 and there is no longwave included
(Response) Thank you for pointing out this important issue in our data. The observation of downward longwave radiation (Ld) is carried out by standard and well calibrated sensor, and regularly maintained by professional meteorological company (Climatec, inc. http://www.weather.co.jp/). However, observation of Ld encountered several instrumental problems since 2000. Until now, we are still trying to recover the missing/error data. Therefore, we did not publish all the observed Ld data during our initial submission. As suggest by the reviewer, we newly release the reliable daily observed Ld data from 2002 to 2006 which are included in the CRiED database (Ver. 1.1, http://www.ied.tsukuba.ac.jp/∼hojyo/archives1.1/yearly/). For the missing/error data, a reliable method established by Kondo and Xu (1997) was employed to estimate the daily Ld. We would like to release both of the reliable observed Ld and calculated Ld data in the supplement data file in section 2.2.

6. P9 Chapter 3 ==> maybe you should also talk about the different available versions here. What is the difference between version 1.0, 1.1 and 2 etc.

(Response) Thank you for your constructive suggestion. It is very important to introduce this database clearly. So, a description of the differences among version 1.0, 1.1 and 2.0 was added to the Chapter 3 according to the reviewer's recommendation to the text. [P9 L18-19, P10 L1-9] "This database includes three versions: Ver. 1.0, 1.1 and 2.0. For Ver. 1.0, the data were collected in integer data format following a former system standard, which applied to the observed data until April 2003. Then, the new system was started from May 2003 and the data set was updated as Ver. 1.1. The data quality is guaranteed by the consistent quality control of all raw observation data. The quality control includes removing error data due to instrumental problems, and missing data caused by observed values out of the specified range (http://www.ied.tsukuba.ac.jp/yosoku/terc/). The data format in Ver. 1.1 was established in accordance with Asanuma et al., 2004. The Ver. 2.0 is the newest version, which is a comprehensive version contains both of the Ver. 1.0 and Ver. 1.1, for the purpose of improving data reliability by performing quality evaluation and quality control. Ver. 2.0 has two main sections. The first section is composed by the hourly, monthly and annual average values with highly consistent quality control, from August 1981 to December 2005. The other one is composed by the raw data, which include data in time frequency of 30min, 60min, 24hour, and 10sec from 2003 to the present (http://www.ied.tsukuba.ac.jp/yosoku/kansoku/rawdata/)." [P9 L18-19, P10 L1-9]

7. P11 Line 16 ==> you talk about daily observed values – please clarify here, that it is daily averages (e.g. temperature) and daily sums (e.g. precipitation?) respectively.

(Response) Thank you for pointing out the important aspect in data treatment. In this study, the daily observed values are estimated from the hourly data. For precipitation, the daily value is the accumulated value based on hourly observation. For the other parameters, the daily value is the average value based on hourly observation. Since the data treatment is consistent and general, we add the description text into P9 L7. Text was added to the context as followed: "For most of the parameters, the daily values are average values from hourly data, except precipitation daily value is the accumulation from hourly data. " [P9 L7-8]

8. P12 Line 10 ==> you state here that the average annual precipitation is 3122 mm/a, and in Figure 5 that seems to be true. BUT on page 4 you say that the region has a long-term annual average precipitation of 1200-1600 mm, and in the 2004 file e.g. the sum of all hourly values for that year amounts to around 1520 mm. So the 3122 mm average, and all values in Fig. 5, seem way too high.

(Response) We appreciated reviewer for carefully evaluating our data. As the reviewer mentioned, the statement of L20, Page 12 L20: "The 37-year average value of precipitation is 3122.1 mm per year" is not correct. We carefully checked the data again, the average value is 1183.8 mm per year. Furthermore, we checked the annual average value from JMA (http://www.data.jma.go.jp/obd/stats/etrn/view/annually_s.php?prec_no=40&block_no=47646&year=&month=&day=&view the average value from 1981 to 2017 is 1259.139 mm/a (3.4497 mm/day). The regression between CRiED and Tateno_JMA was shown in Figure 2. Based on this relation, the daily value could be regressed as: CRiED = 0.8529 * Tateno_JMA(daily value) + 0.1794 (shown as P11, Figure 2.). So the regressed value should be 1139.4 mm/a for CRiED. So, the revised annual precipitation of 1183.8 mm/a is reasonable. To modify this mistake, we rephrase our statement to the following and present the corrected value in the manuscript: "The 37-year average value of precipitation is 1183.8 mm per year," [P12 L20]

9. P15 Description of Figure 4 ==> Why do you mention soil and dew point temperature, but not air temperature, which is also included in Figure 4? Also other parameters are not mentioned here, why?

(Response) The original description was incomplete. We modified the description of Figure 4 and the text was added to the maintext as following: "Daily observed values of the air temperatures for all layers, maximum, minimum and mean air temperature at a height of 1.6 m, soil and dew temperatures for all layers, the precipitation, air pressure, humidity, wind speed, longwave radiation, solar radiation, net radiation, sensible heat flux and the soil heat flux at the EDP site from 1981 to 2017." [P16 Figure 4.]

Technical corrections 1. P1 Line 13 ==> If you write "depth", normally the numbers are not negative. I would remove the "-" [minus] signs when you write "at depths of -0.02 m, . . ."

(Response) We thank the reviewer for pointing this out. The minus signs were removed accordingly. [P1 L13-14]

2. P4 Line 4 ==> in the lat/long the 36.0âŮę can be reduced to 36âŮę (remove ".0")

(Response) We thank the reviewer for pointing this out. We have changed the coordinate "36.0°06'35" N" to "36°06'35" N". [P4 L4]

3. P4 Figure 1 ==> Please enhance the size of the font a bit, as the X and Y axis is not easily readable

(Response) We thank the reviewer for pointing this out. The font size of X and Y axis were enlarged in the revised Figure 1. [Figure 1. P4]

4. P5 Line 1-3 ==> Please use capital first letter for the genus, e.g. Imperata cylindrica (no I in cylindrica!), Andropogon virginicus (Andropogon, not Audropogon!) etc.; Lespedeza cuneata is with a at the end, not e (probably all automatic spell check errors)

(Response) We thank the reviewer pointing out the mistakes and appreciate his/her broad knowledge in Ecology. We modified the spell and use the first capital letter for the genus. The revised text to the context as following: "The vegetation is naturally grown C3 and C4 vegetation, such as Imperata cylindrica, Andropogon virginicus, Miscanthus sinensis as C4, and Solidago altissima, Artemisia princeps, Lespedeza cuneata, Lespedeza pilosa, Equisetum arvense, Festuca arundinacea, Potentilla freyniana, Lysimachia clethroides as C3." [P5 L1-3]

5. P5 Table 1 ==> The date format is a little weird here. I would maybe use ISO format (2015-12 or 1981-08 etc.)

(Response) According to the reviewer's suggestion, we improved the date format as ISO format. The modified context could be found in Table 1. [Table 1. P5-6]

6. P18 Line 3 ==> as you give units for all parameter, you should also list "%" for relative humidity

(Response) We thank the reviewer for pointing this out. The unit "%" is added to the description of Figure 5. [P19 L3]

7. References ==> you list three references as "in review" (Godsey, Kosmos, Makarieva). All three have been published in the meanwhile, so you should update the references

(Response) We updated the references in the context accordingly: "Godsey, S. E., Marks, D., Kormos, P. R., Seyfried, M. S., Enslin, C. L., Winstral, A. H., McNamara, J. P., and Link, T. E.: Eleven years of mountain weather, snow, soil moisture and streamflow data from the rain–snow transition zone – the Johnston Draw catchment, Reynolds Creek Experimental Watershed and Critical Zone Observatory, USA, Earth Syst. Sci. Data, 10, 1207-1216, https://doi.org/10.5194/essd-10-1207-2018, 2018. " [P20 L22-25] "Kormos, P. R., Marks, D. G., Seyfried, M. S., Havens, S. C., Hedrick, A., Lohse, K. A., and Sandusky, M.: 31 years of hourly spatially distributed air temperature, humidity, and precipitation amount and phase from Reynolds Critical Zone Observatory, Earth Syst. Sci. Data, 10, 1197-1205, https://doi.org/10.5194/essd-10-1197-2018, 2018." [P21 L21-23] "Makarieva, O., Nesterova, N., Lebedeva, L., Sushansky, S.: Water balance and hydrology research in a mountainous permafrost watershed in upland streams of the Kolyma River, Russia: a database from the Kolyma Water-Balance Station, 1948–1997, Earth Syst. Sci. Data, 10, 689-710, https://doi.org/10.5194/essd-10-689-2018, 2018." [P22 L1-3]

Finally, comments from the anonymous reviewer are helpful in improving our manuscript. So, we expressed our gratitude to ACKNOWLEGEMENT: "We thank anonymous reviewer for the thoughtful and constructive comments, which helped improve the quality of this work." [P20 L10-11]

Please also note the supplement to this comment:
https://www.earth-syst-sci-data-discuss.net/essd-2018-58/essd-2018-58-AC1-supplement.zip

---

## Author Comment (AC2) · 10 Sep 2018

One more remark: sorry I forgot the following: In the abstract you cite "Konto and Xu (1997)", but that should probably mean "Kondo and Xu (1997)"

(Response) We thank the reviewer for pointing this out. We modified the text as followed: "Kondo and Xu (1997)" [P1 L20]

Furthermore, we added two references wrote by Xu et al., which helps readers to understand this method well. The context added to the manuscript are: "This method was successfully applied to Tibetan Plateau (Xu and Haginoya, 2001, Xu et al., 2005)." [P7 L5-6] The references to the context as followed: "Xu, J., Haginoya, S.: An Estimation of Heat and Water Balances in the Tibetan Plateau, Journal of the Meteorological Society of Japan, 79(1B), 485–504, doi: 10.2151/jmsj.79.485, 2011." [P23 L14-15] "Xu, J., Haginoya, S., Masuda, K., Suzuki, R.: Heat and Water Balance Estimates over the Tibetan Plateau in 1997-1998, Journal of the Meteorological Society of Japan, 83(4), 577–593, doi: 10.2151/jmsj.83.577, 2005." [P23 L16-17]

---

## Referee Comment (RC3) · Anonymous Referee #2 · 25 Sep 2018

**General comments** Paper describes a useful long term dataset and I think the authors have put a great deal of work into data management.

I have slight concerns about the doi'd 'Asset' data: http://www.ied.tsukuba.ac.jp/en/edps/database-doi/ which are available as .dat files because .dat files need processing and do not contain header rows; indeed the headers for this data need to be added separately. Why did the authors not publish the data as MS Excel in the same way that the data available from the ESSD supplement (daily data) - opens with no processing and contains a header row. This may just be my preference and still think it's a valuable dataset.

**Specific comments**

I have had to remove "the" in the manuscript where it spoils the flow of words. I have

attached these comments as a supplement pdf.

**Technical corrections** Page 1 Line 10 from a well maintained Line 11 include short-wave radiation, air and dew point temperatures at three elevations, soil temperature Line 12 depths, sensible shortwave, net radiation, air and dew point temperatures at three elevations, soil Line 13 depths, sensible heat flux Line 14 includes four Line 15 presented here. Monthly Line 17 We validated the data by Line 22 and percent bias Line 23 values show a positive trend in precipitation Line 24 over the past 37 years with a negative trend detected for wind speed Page 2 Line 1 historical Line 3 from a well-maintained Line 11 for estimating atual Line 13 the effect of Line 19 - 21 Consider revising sentence. I don't understand what you mean Line 21 the understanding Line 22 Yamanaka et al. (2005) carried out quality control of the data Line 23.Validation of the water Line 25 Consider revising sentence. I don't understand what you mean Line 26 In addition, latent heat flux was assessed by Line 27 with flux behaviour Line 28 into temporal variation, and an assessment of measurement accuracy Line 29 not only above ground, but also within the soil layers, have Line 31 observations of soil temperature Page 3 Line 1 A wide variety of studies on ecology and vegetation were conducted Line 4 (1989) estimated turbulent fluxe using the eddy-covariance method to investigate the effect of forest and vegetation Line 6 Being a well maintained observation site, the grassland provides Line 9 conducted experiments to investigate the biomass and Page 4 Line 6 site consist of a grass covered circular field, 160m in diameter at an altitude of Line 7 at a height of 30 metres Figure 1 I can't read the axis titles or legend on (d) Line 17 top few metres – can you be more exact here? Page 5 Line 1, 2, 3 vegetation names need a capital for first word e.g. Imperata cylindrica and please check spellings for accuracy; please retain italics Lines 4 and 5 The similarity of grass species, depth, and leaf-area index (LAI) were confirmed annually by two different surveys. Line 4 to 9 – are these the two surveys. This is a bit confusing because it states each year on line 4 and then 'two years later' on line 7. Consider revising paragraph to make it clearer. Line 10 The grass is mown twice per year (summer and winter since 2006); dead plants and grass clippings are redistributed.

[Figure]

Table 1. would be best viewed on one page please Page 6 Line 15 The data are freely available for download from the Center for Research in Isotopes and Environmental Dynamics (CRiED) website (http://www.ied.tsukuba.ac.jp/en/edps/database-doi/) (formerly known as TERC) as hourly, monthly and annual summaries. Since 2003, the temporal resolution is at 10 second, 30 minutes, 60 m and 24 hour intervals. When calculating averaged data (Asanuma et al., 2004) at least 24 records at 30 m were required. Readings with less than 20 records were discarded and data with between 20 and 24 records were annotated as incomplete (Ohba and 20 Yamanaka, 2007). In addition to the missing data, dates of equipment maintenance, construction and mowing information are recorded in the maintenance log (http://www.ied.tsukuba.ac.jp/yosoku/kansoku/hojyo$_{log}$/).$Page 9. Line 2. Data quality was checked for Line 19 including short w$

In the data provided as ESSD Supplement: Column heading spelling mistake: Air tempearture should be Air$_{temperature}$

$Please also note the supplement to this comment:$
$https://www.earth-syst-sci-data-discuss.net/essd-2018-58/essd-2018-58-RC3-supplement.pdf$
* * *
[Figure]

**Supplement:**

Comment on "A database of water and heat observations over grassland in the north-east of Japan" by Wenchao Ma et al.

Anonymous Referee #2

**General Comments**

Paper describes a useful long term dataset and I think the authors have put a great deal of work into data management.

I have slight concerns about the doi'd 'Asset' data: http://www.ied.tsukuba.ac.jp/en/edps/database-doi/ which are available as .dat files because .dat files need processing and do not contain header rows; indeed the headers for this data need to be added separately. Why did the authors not publish the data as MS Excel in the same way that the data available from the ESSD supplement (daily data) - opens with no processing and contains a header row? This may just be my preference and still think it's a valuable dataset.

**Specific comments**

I have had to remove 'the' in the manuscript where it spoils the flow of words.

**Technical corrections**

Line 10 from a well maintained

Line 11 include  shortwave radiation,  air and dew point temperatures at three elevations,  soil temperature

Line 12 depths,  sensible shortwave, net radiation,  air and dew point temperatures at three elevations,  soil

Line 13  sensible heat flux

Line 14  four

Line 15 presented here. Monthly

Line 17 We  validated the data  by

Line 22 and  percent bias

Line 23 values  show a positive trend in  precipitation

Line 24 over the  past 37 years

Line 25 detected for  wind speed

Line 1  historical

Line 3 from a well-maintained

Line 12 for estimating

Line 13 effect

Line 19 - 21 By assessing the observed evaporation in 2001, the results estimated from the Penman, energy budget eddy covariance, and energy-balance Bowen ratio methods were presented (Yubasaki et al., 2005), which improved understanding of the variation in evaporation from a conversion in the fraction of pasture at the site into turf. Consider revising sentence. I don't understand what you mean

Line 21 the understanding

Line 22 Yamanaka et al. performed carried out quality control of the data

Line 23. The vValidation of the water budget was

Line 25 where a model for the estimation of the precipitation on the grassland of the EDP department was developed, showing a good adaptability with a model taking into account the canopy, stem and evapotranspiration components based on observations from the EDP database. Consider revising sentence. I don't understand what you mean

Line 26 The In addition, latent heat flux was also assessed

Line 27 with the flux behaviour

Line 28 into the temporal variation, and an the assessment of the measurement accuracy

Line 29 not only above the ground

Line 31 observations of the soil

Line 1 A wide variety of studies on ecology and vegetation were broadly conducted

Line 4 (1989) estimated the turbulent fluxes using the eddy-covariance method for assessing the effect of the to investigate the effect of

Line 6 Being a well maintained observation site, the grassland gives provides

Line 9 conducted experiments for investigation of to investigate the

Line 5 site consist of a grass covered circular field 160m in diameter at an altitude of

Line 7 of 30-m height at a height of 30 metres

Figure 1 I can't read the axis titles or legend on (d)

Line 17 top few metres – can you be more exact here?

Line 1, 2, 3 vegetation names need a capital for first word e.g. *Imperata cylindrica* and please check spellings for accuracy

Lines 4 and 5 The similarity of grass species, and their depth, as well as their and leaf-area index (LAI), were also explicitly confirmed annually each year by two different surveys.

Line 4 to 9 – are these the two surveys. This is a bit confusing because it states each year on line 4 and then 'two years later' on line 7. Consider revising this whole paragraph to make it clearer.

Line 10 Since 2006, the grass has been mown twice each year ( summer and winter);  dead plants and grass clippings were redistributed.

Table 1. would be best viewed on one page please

Line 14 Data collected from supersonic anemometer–thermometers  was used to obtain

Line 15  The data are freely available for download from  the Center for Research in Isotopes and Environmental Dynamics (CRiED) website (http://www.ied.tsukuba.ac.jp/en/edps/database-doi/) (formerly known as TERC) as hourly, monthly and annual summaries. Since 2003, the temporal resolution is at 10 second, 30 minutes, 60 m and 24 hour intervals.

When calculating averaged data (Asanuma et al., 2004) at least 24 records at 30 m were required. Readings with less than 20 records were discarded and data with between 20 and 24 records were annotated as incomplete (Ohba and 20 Yamanaka, 2007). In addition to the missing data, the dates of equipment maintenance, construction and mowing information are recorded in the maintenance log (http://www.ied.tsukuba.ac.jp/yosoku/kansoku/hojyo_log/).

Page 9.

Line 2. Data quality was

Line 19  shortwave radiation

Line 9  differs slightly

Differences in shortwave radiation are mainly governed by solar radiation, whereas absorption and reflection may be caused by atmospheric conditions (clouds)

Line 12  wind speed

**In the data provided as ESSD Supplement:**

Column heading spelling mistake: **Air temperature** should be **Air_temperature**

---

## Author Comment (AC3) · 11 Oct 2018

Point-by-point responses to the comments of Anonymous Reviewer #2

Below, we have outlined our responses and the changes we have made in the revised manuscript and the Supplementary Information. In this document, the reviewers' comments are shown in bold italic face, our responses in blue and revisions in red. We have similarly highlighted all the changes in red in the revised Manuscript.

———————————————

"Paper describes a useful long term dataset and I think the authors have put a great deal of work into data management." (Response) We thank the reviewer for his/her recommendation!

[Figure]

I have slight concerns about the doi'd 'Asset' data: http://www.ied.tsukuba.ac.jp/en/edps/database- doi/ which are available as .dat files because .dat files need processing and do not contain header rows; indeed the headers for this data need to be added separately. Why did the authors not publish the data as MS Excel in the same way that the data available from the ESSD supplement (daily data) - opens with no processing and contains a header row? This may just be my preference and still think it's a valuable dataset." (Response) We thank the reviewer for the comments. We understand that it is inconvenient for using .dat files. However, this database is used by many individuals and institutes since decades ago, which is still under way. Most of the usage were set as automatically collecting the real-time data dealing with the .dat files. So, it will cause troubles for these existing users if the data format was changed. Although it is our duty to provide convenient for these existing users, we are seriously considering how to provide an easier way to serve more users. For example, we are submitting a clear and carefully checked Excel file in this work for wider application.

Specific comments I have had to remove 'the' in the manuscript where it spoils the flow of words. (Response) Thanks very much for your valuable suggestions. Those misused "the" and syntax error were modified according to the reviewer's guidance. Please check the following revisions: Page 1: Line 10 "…from a well maintained …" [P1 L10] Line 11, 12 "…observations include shortwave radiation, net radiation, air and dew point temperatures at three elevations, soil temperature…" [P1 L11-12] Line 13 "…sensible heat flux,…" [P1 L13] Line 14 "…includes four temporal resolutions…" [P1 L14] Line 15 "Monthly and annual…" [P1 L15] Line 17 "We validated the data by…" [P1 L16-17] Line 22 "…and percent bias…" [P1 L22] Line 23 "…annually averaged values show a positive trend in precipitation,…" [P1 L22] Line 24 "…over the past 37 years,…" [P1 L23] Line 25 "…detected for wind speed,…" [P1 L24] Page 2 Line 1 "…recording historical climatic variation…" [P1 L30] Line 3 "…from a well-maintained grassland…" [P2 L2] Line 12 "…for estimating actual evapotranspiration…" [P2 L10-11] Line 13 "…with the effect of stemflow and vegetation…" [P2 L12-13] Line 26 "…In addition,

latent heat flux was assessed…" [P2 L24] Line 27 "…with flux behavior…" [P2 L24] Line 28 "…into temporal variation, and an assessment of measurement accuracy…." [P2 L25] Line 29 "…not only above ground,…" [P2 L26] Line 31 "…observations of soil…" [P2 L28] Page 3 Line 1 "A variety of studies on ecology and vegetation were conducted…" [P2 L32] Line 4 "…to investigate the effect of…" [P3 L1] Line 6 "Being a well-maintained observation site, the grassland provides…" [P3 L3] Line 9 "…conducted experiments to investigate the…" [P3 L6] Page 4 Line 5 "…site consists of a grass covered circular field 160 m in diameter at…" [P4 L2] Line 7 "…at a height of 30 meters…" [P4 L3] Page 5 Lines 4, 5 "…similarity of grass species, depth, and leaf-area index (LAI), were confirmed annually by…" [P4 L21-22] Line 10 "Since 2006, the grass has been mown twice each year (summer and winter); dead plants and grass clippings were redistributed." [P5 L5-6] Page 6 Line 14 "Data collected from super-sonic anemometer–thermometers was used to obtain heat …" [P6 L11] Page 9 Line 2. "…data quality was checked…" [P8 L21] Line 19 "…including shortwave radiation,…" [P10 L5] Page 10 Line 9 "…RH differs slightly,…" [P10 L14] Line 12 "…except wind speed and precipitation." [P10 L16]

Page 2 Line 19 - 21 By assessing the observed evaporation in 2001, the results estimated from the Penman, energy budget eddy covariance, and energy-balance Bowen ratio methods were presented (Yubasaki et al., 2005), which improved understanding of the variation in evaporation from a conversion in the fraction of pasture at the site into turf. Consider revising sentence. I don't understand what you mean Line 21 the understanding (Response) Thank you for underlining this deficiency. We have re-written this part according to the Reviewer's suggestion. The revised text to the context as following: "By evaluating the observations in 2001, Yubasaki et al., (2005) tested a reduction factor for pasture can be used to the turf site by comparing the evapotranspiration estimated by Penman, energy-budget eddy covariance, and energy-balance Bowen ratio methods." [P2 L18-20]

Line 22 Yamanaka et al. performed carried out quality control of the data (Response)

We are grateful for the suggestion. Because of the reference cited here is the "Saito, M., Yamanaka, T.: Analysis of Long-term Evapotraspiration Data Observed by Weighing Lysimeter and Its Quality Control, Bulletin of the TERC, the University of Tsukuba, 6, 53–62, doi: 10.15068/00147122, 2005." [P22 L17-18], so the revised text to the context as following: "Saito and Yamanaka (2005) carried out a quality control of the data, and analyzed the evapotranspiration data observed with a weighing Lysimeter between 1981 and 2002, with the results of the data quality summarized." [P2 L20-22]

Line 23. The vValidation of the water budget was Line 25 where a model for the estimation of the precipitation on the grassland of the EDP department was developed, showing a good adaptability with a model taking into account the canopy, stem and evapotranspiration components based on observations from the EDP database. Consider revising sentence. I don't understand what you mean (Response) We thank the reviewer for pointing this out. We reorganized the sentence and revised text to the context as following: "In 1983, a model was developed by Tase and Majima for estimating precipitation under the influence of interception. This model showed a good adaptability with the canopy, stem and evapotranspiration components." [P2 L22-23]

Page 4 Figure 1 I can't read the axis titles or legend on (d) (Response) Thanks for this thoughtful comment. The font size of X and Y axis were enlarged in the revised Figure 1. [P4]

Line 17 top few metres – can you be more exact here? (Response) We thank the reviewer for pointing this out. We take a further literature review about this, and add this literature in the reference. The top 1 or 2 metres was filled by loam and volcanic ash soil. Below the first layer, is a clay layer with thickness of 4 or 5 metres. So, we revised the text as following: "The observational site was artificially filled with loam and volcanic ash soil in the top of 1 $\sim$ 2 meters, and clay layer with thickness of 4 $\sim$ 5 meters underneath (Sakura, 1977)." [P4 L13-14] We updated the references in the context accordingly: "Sakura, Y.: Miscellaneous, âĚč Water balance observation facility, Bulletin of the ERC, the University of Tsukuba, 1, 87–90, 1977." [P21 L19-20]

Page 5 Line 1, 2, 3 vegetation names need a capital for first word e.g. Imperata cylin-drica and please check spellings for accuracy (Response) Thank you for this valuable comment. The revised text to the context as following: "The vegetation is naturally grown C3 and C4 vegetation, such as Imperata cylindrica, Andropogon virginicus, Miscanthus sinensis as C4, and Solidago altissima, Artemisia princeps, Lespedeza cuneata, Lespedeza pilosa, Equisetum arvense, Festuca arundinacea, Potentilla frey-niana, Lysimachia clethroides as C3." [P4 L19-21]

Line 4 to 9 – are these the two surveys. This is a bit confusing because it states each year on line 4 and then 'two years later' on line 7. Consider revising this whole paragraph to make it clearer. (Response) Thank you for pointing out this deficiency. Actually, they are two unrelated surveys, so we deleted the ambiguous 'two years later'. The revised text to the context as following: "Another survey was carried out between 2000 and 2002 to directly measure the LAI and height, with similar results found as before," [P5 L2-3]

Table 1. would be best viewed on one page please (Response) Thanks for your sug-gestion. We have modified the Table 1 within Page 5. [P5]

Page 6 Line 15 These observational data The data are freely available for download from open to the public and are the Center for Research in Isotopes and Environ-mental Dynamics (CRiED) website (http://www.ied.tsukuba.ac.jp/en/edps/database-doi/) (formerly known as TERC) as hourly, monthly and annual summaries. Since 2003, the temporal resolution is at 10 second, 30 minutes, 60 m and 24 hour in-tervals. through our website ()", which is renewed updated every minute When calculating averaged data (Asanuma et al., 2004) at least 24 records at 30 m were required. Readings with less than 20 records were discarded and data with between 20 and 24 records were annotated as incomplete (Ohba and 20 Ya-manaka, 2007). In addition to the missing data, the dates of equipment mainte-nance, construction and mowing information are recorded in the maintenance log (http://www.ied.tsukuba.ac.jp/yosoku/kansoku/hojyo_log/). (Response) We appreciated the reviewer very much for organizing the paragraph more legible and easy to read. The revised text to the context as following: "The data are freely available download from the Center for Research in Isotopes and Environment Dynamics (CRiED) website (http://www.ied.tsukuba.ac.jp/en/edps/database-doi/) (formerly known as TERC) as hourly, monthly and annual summaries. Since 2003, the temporal resolution is at 10-second, 30-minute, 60-minute and 24 hour intervals. When calculating averaged data (Asanuma et al., 2004) at least 24 records at 30 minutes were required. Readings with less than 20 records were discarded (marked with "***" in the supplement data file) and data with between 20 and 24 records were annotated as incomplete (Ohba and Yamanaka, 2008; marked with "*" in the supplement data file). In addition to the missing data, the dates of equipment maintenance, construction and mowing information are recorded in the maintenance log (http://www.ied.tsukuba.ac.jp/yosoku/kansoku/hojyo_log/)." [P6 L12-20]

Page 10 Differences in shortwave radiation are mainly governed by solar radiation, whereas absorption and reflection may be caused by atmospheric conditions (clouds) (Response) We thank the reviewer for pointing this out. We have made correction accordingly. The revised text to the context as following: "Differences in shortwave radiation between the EDP and Tateno are mainly governed by solar radiation, whereas absorption and reflection may be caused by atmospheric conditions (clouds)." [P10 L14-16]

In the data provided as ESSD Supplement: Column heading spelling mistake: Air temperature should be Air_temperature (Response) Thank you for this valuable comment. The mistake of the column heading spelling was modified accordingly.

Finally, comments from the anonymous reviewer are helpful in improving our manuscript. So, we expressed our gratitude to ACKNOWLEGEMENT: "We thank two anonymous reviewers for their thoughtful and constructive comments, which helped improve the quality of this work." [P19 L10-11]

Please also note the supplement to this comment:
https://www.earth-syst-sci-data-discuss.net/essd-2018-58/essd-2018-58-AC3-
supplement.zip

———————————————————————